# Chaining and the temporal dynamics of scientists' publishing behaviour

**Emmy Liu** [1]*, **Yang Xu** [2]

**1** Language Technologies Institute, School of Computer Science, Carnegie Mellon University, Pittsburgh, PA, United States of America, **2** Department of Computer Science, Cognitive Science Program, University of Toronto, Toronto, Canada

* mengyan3@cs.cmu.edu

## Abstract

Scientific progress, or scientific change, has been an important topic in the philosophy and history of science. Previous work has developed quantitative approaches to characterize the progression of science in different fields, but how individual scientists make progress through their careers is not well understood at a comprehensive scale. We characterize the regularity in the temporal dynamics of scientists' publishing behavior with computational algorithms that predict the historical emerging order of publications from individual scientists. We discover that scientists publish in ways following the processes of chaining that mirror those observed in historical word meaning extension, whereby novel ideas emerge by connecting to existing ideas that are proximal in semantic space. We report findings for predominant exemplar-based chaining that reconstructs the emerging order in the publications of 1,164 award-winning and random-sampled scientists from the fields of Physics, Chemistry, Medicine, Economics, and Computer Science over the past century. Our work provides large-scale evidence that scientists across different fields tend to share similar publishing behavior over time by taking incremental steps that build on their past research outputs.

## Introduction

The process of scientific inquiry can follow a winding path that varies across individuals. Isaac Newton explored optics in his earlier career but later studied the laws of motion and gravity [1]. Alan Turing at his earlier age worked on computability theory but later contributed to machine intelligence and the chemical basis of morphogenesis [2]. The problem of scientific change, or the progression of science through generations of scientists, has garnered much attention in the philosophy and history of science. However, whether there is regularity in how individual scientists make progress through time has not been studied comprehensively across scientific fields.

Scientific change has been an important topic in the philosophy of science. Kuhn suggests that science develops in cyclic periods of normal science and revolutionary science [3]. Normal science depends on similarities to exemplars, or canonical instances of scientific work that

**Data Availability Statement:** All data and code are available at https://github.com/nightingal3/metascience/tree/master.

**Funding:** YX was supported by NSERC Discovery Grant RGPIN-2018-05872 (https://www.nserc-crsng.gc.ca/professors-professeurs/grants-subs/

dgigp-psigp_eng.asp), SSHRC Insight Grant #435190272 (https://www.sshrc-crsh.gc.ca/funding-financement/programs-programmes/insight_grants-subventions_savoir-eng.aspx), and Ontario Early Researcher Award #ER19-15-050 (https://www.ontario.ca/page/early-researcher-awards) The funders played no role in study design, data collection and analysis, or preparation of the manuscript.

**Competing interests:** The authors have declared that no competing interests exist.

scientists base their hypotheses on. However, anomalies challenging the existing paradigm may build up and give rise to a period of revolutionary science, also known as a paradigm shift. Kuhn's view resonates with Popper's work emphasizing falsifiability as an avenue for the advancement of science [4], and Fayerabend's incommensurability thesis arguing that successive scientific paradigms may have entirely different conceptual structures and language uses that are incompatible with each other [5]. Other scholars have suggested that paradigm shifts can happen gradually during periods of normal science, and the distinctions between normal science and revolutionary science may be blurry [6]. The philosophical inquiries of scientific change have focused on explaining scientific progress at a macro scale, but they tend to under-explore mechanisms in the scientific trajectories of individual scientists.

Parallel studies of scientific change have been conducted in the history of science and cognitive science. For instance, existing work has investigated the transition from Newtonian classical mechanics to the theory of relativity and quantum mechanics [7], and the effects of new instruments and methods in physics and chemistry [8]. Transition from one conceptual system to another involves an individual scientist noticing contradictions and inconsistencies within the old conceptual system, and constructing new representations of the scientific concepts. This line of work in cognitive science focuses on individuals, but it typically relies on anecdotal case studies.

Recent work has taken more scalable and quantitative approaches to studying scientific progress, but tends to focus on the evolution at the level of a scientific field. Existing work has explored scientific topic shifts, networks of scientific collaboration, as well as the connections among people, topics, methods, and team scale in scientific research [9–16]. Recent work has also explored the role of atypicality and outliers in scientific publications, such as how "sleeping beauty" papers that became highly influential over the long term at the expense of short-term citations, or understanding the factors underlying "outlier patents" [17, 18]. However, focusing on outliers and exceptional papers might obscure the incremental steps by which such papers are generated, since exceptional or unusual papers from a community perspective could result from the logical continuation of a line of research from an individual perspective. The problem we pursue here is whether simple computational algorithms focused on characterizing fine-grained temporal dynamics of scientists' publication behavior might reveal general patterns in scientific progress made by individual scientists in different fields.

Although scientific inquiry over time can vary greatly across individuals, we hypothesize that there should be regularity in scientists' publication behavior through time that reflect common principles of human cognition. Our approach is to uncover this regularity by testing cognitively inspired algorithms against the historical records of scientists' research outputs as they emerged over time in different fields.

Our proposal is grounded in the cognitive theory of chaining [19–22], and we use the existing computational cognitive algorithms that have been shown to capture properties of chaining in the context of word meaning extension [22–25]. More specifically, chaining refers to a continuous and incremental process of meaning growth whereby novel ideas connect to existing ideas that are semantically similar, hence forming chain-like structures over time. Existing work has formulated chaining as probabilistic algorithms or models of categorization and applied these formal models to predict the historical growth of word meaning in a variety of domains including container names, numeral classifiers, adjectives, and verbs [22–27].

We believe that the general process of scientific progress might mirror the process of word meaning extension, whereby a scientist conceives of a new idea by linking to existing ideas that are closely related. Importantly, previous work has also suggested that certain forms of chaining are preferred or more cognitively natural than other forms, and recent findings suggest that chaining can be best understood as an exemplar-based process [22, 24, 25]. The exemplar

model of chaining is rooted in the exemplar theory of categorization, also known as the General Context Model [28]. This model has been demonstrated to predict the historical meaning extensions of numeral classifiers [22], adjectives [24], and verbs [25].

Here we investigate whether similar chaining mechanisms might account for the historical emergence of scientific publications from individual scientists. Independent work in mathematical modeling has proposed that similar processes may underlie the emergence of novelties [29]. However, to our knowledge there has been no formalization or comprehensive evaluation of chaining in the temporal dynamics of scientists' publishing behavior over time. We focus on investigating three related issues: 1) whether there is systematic predictability in the historical order of emergence of scientists' papers; 2) how the exemplar-based chaining account fares against alternative competing accounts including the prominent prototype-based view that predicts growth to stem from a central core [30, 31]; 3) whether mechanisms of chaining are domain-general or differ across scientific fields, or between award-winning (or prominent) and randomly-sampled (control) scientists.

To evaluate our proposal at scale, we collected a large set of historical publication data from scientists who vary along two dimensions: 1) scientists in different scientific fields, including Physics ($n = 168$), Chemistry ($n = 120$), Medicine ($n = 151$), Economics ($n = 74$), and Computer Science ($n = 69$); 2) award-winning scientists ($n = 582$) who have made prominent contributions in these fields, including Nobel Laureates and Turing Prize winners, versus randomly sampled scientists ($n = 582$ in total). Within each field and its corresponding prize category (e.g., Physics and Nobel Prize for Physics), we extracted data exhaustively from all prize-winning scientists since the 20th century, and we constructed the random control group by sampling scientists from the same field and matching the group size to that within the prize category. For each scientist, we collected his or her publications throughout history where the records were available, and we timestamped each publication by the year that the work was published. We describe details of the dataset and its processing in *Materials and Methods*.

In our analysis, we applied chaining models to the following prediction task: Given a scientist's publications up to time *t*, predict which of the yet-to-emerge publications will more likely to appear at the next time step *t*+ 1. This prediction task is similar to the problem formulated in previous work that examines regularity in the historical emergence of word meanings [23]. Here instead we use this predictive analysis to explore shared patterns that underlie the emergence of scientific publications. If the exemplar-based chaining reflects a general and preferred mechanism in the emergence of scientific papers, we expect this model to best account for data across the 5 scientific fields, and in both the groups of prominent and random-sampled scientists.

## Computational framework

We formulate the computational problem by considering the set of papers $S_t$ a scientist has published by time *t*, and then applying models of chaining incrementally over time to infer the probability of a yet-to-emerge paper $x^*$ from the same scientist actually being published at the next available time $t + 1$. Our analysis treats each year as an individual timestep, so papers published within the same year are considered to emerge at the same time. We represent the publications of a scientist as a point cloud, where each point denotes a publication. As time unfolds, we ask the question whether there is predictability in how this cloud of points emerges over time. We expect papers that emerge at each next point to be generally related to the previous papers that have emerged, and we formalize the temporal dynamics in the emergence of scientific publications using a suite of chaining models grounded in the literature.

Each model of chaining predicts which papers are most likely to emerge at the next timestep based on a softmax function of the similarities of all papers which are yet to emerge, following Luce's choice model [32]:

$$p(x^*|S_t) \sim \frac{exp(f(x^*, S_t))}{\Sigma_{x \in S_t^*} exp(f(x, S_t))}, \tag{1}$$

Here $x^*$ is drawn from the set of papers yet to emerge (or be published) beyond $t$ which we denote as $S_t^*$, and $f()$ specifies a particular model of chaining. As a model predicts through time, $S_t$ is updated with the ground truth papers at each timestep in order to minimize error propagation. Whenever "similarity" is mentioned below, we refer to a standard metric used in measuring psychological distance [28, 33, 34], the exponentiated squared Euclidean distance, between paper embeddings $x_1$ and $x_2$ in a semantic space.

$$sim(x_1, x_2) = \exp(-d(x_1, x_2)^2) \tag{2}$$

We capture the semantic gist of a scientific paper by its content in the abstract section. We choose to represent a paper by the abstract due to its concise and informative character following a procedure adopted commonly in recent computational studies of scientific publications [35, 36]. Specifically, we represent each paper by a sentence embedding of the abstract generated by a model in natural language processing called SBERT [37], that we used the pretrained version of. This sentence embedding represents the semantic content of each abstract as a point in a high-dimensional vector space. Fig 1 provides a 2-dimensional visualization for the semantic space that includes paper embeddings of two prominent scientists in the fields of computer science and chemistry. *data in* S1 Appendix includes analyses that validate the measure we used to represent scientific papers. More specifically, we validate that the SBERT embeddings are able to capture semantic differences among abstracts published in different scientific fields, as well as among the abstracts written by individual scientists within fields.

We consider five models of chaining following the existing literature [22–24]. Each model postulates a plausible yet different cognitive mechanism for how future scientific outputs might relate to or stem from existing ones and constructs a continuous temporal path among the scientific publications of a scientist. Each chaining process is specified by a likelihood function listed in Table 1.

**k-Nearest-neighbor (kNN)** models are commonly used for classification and regression, but in this case we use them to predict which paper from $S_t^*$ is likely to emerge next based on that paper's similarity to the k most similar papers that have already appeared in $S_t$. This model captures the intuition that scientists preferentially publish work that is most similar to any one paper they have published before (e.g., 1NN), or most similar to a group of k papers they have published before (e.g., kNN). We tested values of k from 1 to 20 for each scientist.

**The prototype model** is motivated by Rosch's work in categorization [30]. This model captures the intuition that a scientist's research is organized around a central theme, yet scientists tend to publish in relation to that theme by gradually adjusting it and branching out over time. We take the prototype of a scientist's papers at $S_t$ to be the average of paper embeddings at $S_t$, and define the probability of a paper emerging at time $t+1$ as proportional to its similarity to the prototype at time $t$. This results in a moving average against which new papers are compared.

**The progenitor model** is a static variant of the prototype model in which the prototype is fixed at the first paper to emerge. This model captures the intuition that a scientist started with a base theme and then progressed consistently in that direction by incrementally making

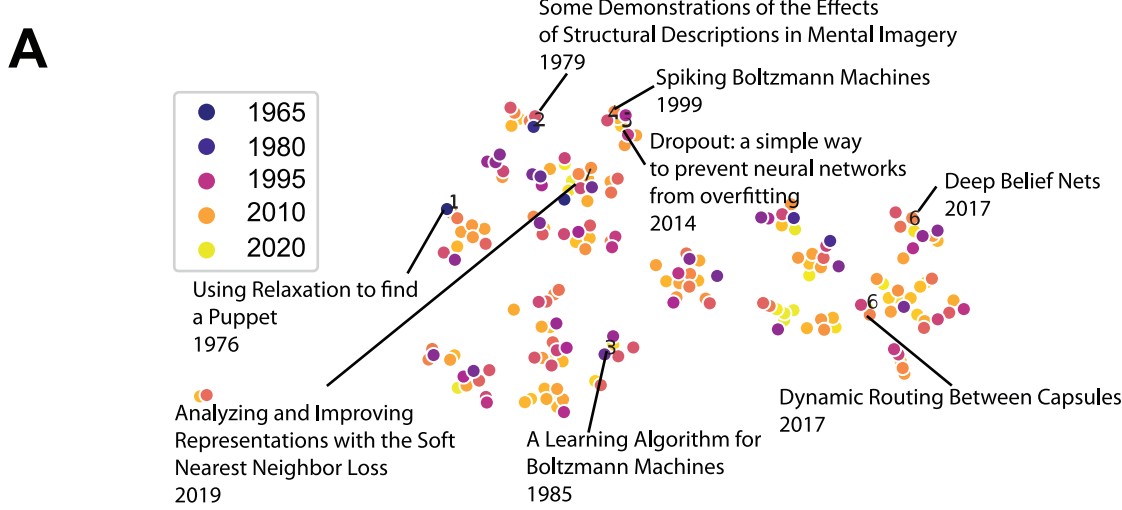

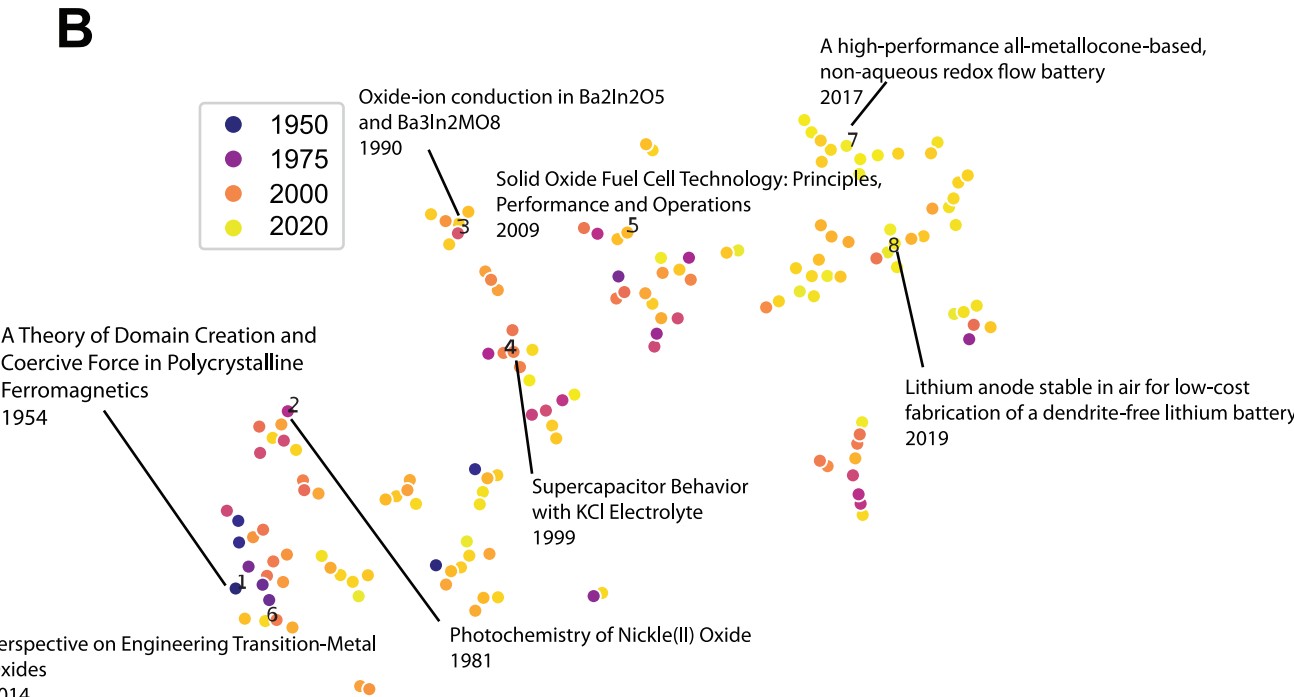

**Fig 1. Visualization of scientific publications over semantic space and time.** (A) Semantic representation of Turing Prize Winner Geoffrey Hinton's published papers throughout his career, with annotations on sampled papers. Each dot represents a paper using BERT-based embedding of its abstract that is projected into 2 dimensions via tSNE dimension reduction [38]. Color gradient indicates year of publication, with more recent papers in warmer colors. Distance between abstract embeddings signifies inverse semantic similarity. (B) Similar information for Nobel Prize Laureate John B. Goodenough. Note that the legend shows the gradation of colours, the latest papers from our dataset were from 2021.

progress toward the theme. In the case where multiple papers emerge in the first year of publication, we take the progenitor to be the average of these papers' embeddings. The probability of a paper emerging at any timestep is then proportional to its similarity only to the progenitor.

**Table 1. Summary of model specifications.** Each model specifies the likelihood of paper $x^*$ to appear at time $t$, given the existing set of papers $S_t$. $d$ denotes Euclidean distance. $C_k(x) \subseteq S_t$ denotes the $k$ closest points to $x$.

| Model class | Likelihood $f(x^*|S_t)$ |
|---|---|
| kNN | $\Sigma_{x' \in C_k(x)} \exp(-d(x, x')^2)$ |
| Prototype | $\exp(-sd(x^*, \text{prototype}(S_t))^2)$ |
| Progenitor | $\exp(-sd(x^*, \text{prototype}(S_0))^2)$ |
| Exemplar | $\Sigma_{x \in S_t} \exp(-sd(x^*, x)^2)$ |
| Local | $\Sigma_{x \in S_t \setminus S_{t-1}} \exp(-sd(x^*, x)^2)$ |

**The exemplar model** is based on work on the exemplar theory [28, 39], which can be viewed as a generalization or soft extension of the kNN models where k is equal to the number of total papers at each timestep but each paper is weighted differentially toward prediction. This model captures the intuition that a scientist tends to publish the next paper that is close to many similar papers they have written before, and the influence of each existing paper is proportional to its similarity to the target paper (i.e., a more similar paper exerts a higher influence). In this model, the probability of a paper emerging at time $t$ is proportional to its similarity to all other papers $S_t$. We consider the most prominent exemplar model known as the Generalized Context Model [28].

**The local model** is a variant of the 1NN model wherein candidate papers are only compared against papers that emerged specifically at the previous timestep. This model captures the intuition that scientists branch out directly from the previous papers they wrote at the previous time step and publish papers close to what they last wrote in terms of time.

Table 1 summarizes the specification of the likelihood function $f()$ for each model class. Together with Eq 1, they specify the probability that each model assigns to a paper emerging at time $t$, given all other papers that have appeared up to that time. Each model class has a single parameter (i.e., model complexity is kept the same for all the chaining models), $k$ in the k-nearest neighbour models and $s$ in the other models. We used sequential model-based optimization implemented in the Python `hyperopt` library to estimate each parameter $s$ [40], and grid searched for $k$ from 1 to 20. The parameter $s$ controls how sharply similarity tapers with increased (Euclidean) distance from the query point.

## Materials and methods

### Data and materials availability

Data and code for replicating the analyses and results reported are deposited as follows: https://github.com/nightingal3/metascience/tree/master.

### Collection and processing of scientific publication data

To capture a diverse and representative range of scientific outputs, we collected the publication data for both award-winning and randomly sampled scientists from five different fields: Physics, Chemistry, Medicine, Economics, Medicine, and Computer Science. We chose the first 4 fields because each of these fields corresponds to an award category of Nobel Prizes, which is one of the most prestigious and long-standing awards for science. We chose computer science because it is a relatively recent scientific field that is distinguished from those of Nobel Prizes, but it nevertheless features the prestigious Turing Award.

We collected all papers available for scientific scholars from DBLP for the Computer Science field, and from Semantic Scholar for the other fields [41, 42]. For each paper, we collected

**Table 2. Sample size of scientists per award category and range of number of publications.**

|  | CS | Chemistry | Economics | Medicine | Physics |
|---|---|---|---|---|---|
| # scientists | 69 | 120 | 74 | 151 | 168 |
| # pub. range | 6–373 | 5–227 | 5–187 | 6–217 | 5–587 |

its title, abstract, year of publication, and coauthor list. We excluded papers from the main analysis if they were missing an abstract or year of publication, and from the authorship analysis if they were missing an abstract, year of publication, or authorship list.

We defined prominent scientists as those who received the highest honour in that scientific field from the inception of the award through to 2020. This was either the Turing Award for the field of Computer Science, or a Nobel Prize for the other fields. For the random group, we randomly sampled scientists from the Semantic Scholar open research corpus labelled with the respective field of study, excluding the set of prominent scientists. We matched the sample size of the random scientists to that of the random scientists. Table 2 summarizes the number of prominent scientists and the range of number of papers published for each field.

## Semantic representation of abstracts

We computationally represented the semantic gist of a scientific paper by embedding its abstract via SBERT [37], which is a common method in natural language processing for representing sentences as a high-dimensional vector. We chose to focus on analyzing abstracts because they provide a compact distillation of the paper contributions while also containing some details on the approach and the background that the paper entails. This procedure follows work on summarization of scientific documents, which often uses the abstract as a target for text summarization [35, 36, 43–45].

Before constructing the abstract embeddings, we used standard preprocessing in natural language processing while also removing textual features common in scientific abstracts that provide irrelevant information. For the standard preprocessing steps, we removed punctuation and stopwords, lemmatized, and removed bracketed words, as these were most often citations. For abstract-specific features, we removed symbols including LATEX, the prefixes "Abstract" and "Summary", as well as subheadings within the abstract summarizing the parts of the paper (e.g., "Background", "Results"). We also excluded any abstracts that were not written in English.

We ran these procedures for each scientist, excluding in turn any scientists that had fewer than 5 papers. For the prominent scientists, to remove unlikely papers, we excluded any papers published after the scientist's death date or fewer than 15 years after their birth date. We converted the dates of publication to timestamps, so that the first paper published would be given timestamp 0, and the second timestamp 1, regardless of their actual publication dates.

We describe the details of a series of tests that evaluate the validity of our semantic representation for the scientific papers in *Supporting Information*.

## Evaluation of chaining models against historical publication data

For each scientist, we calculated the predictive probability of a model based on the sequence of papers produced by that scientist, where the sequence recapitulates the historical emerging order recorded in the years of publications of the papers. The predictive probability of a sequence of papers is the product of the predictive probabilities of the individual papers as they emerged through time: $P = \prod_t p(x^*|S_t)$, where $x^*$ denotes the newly emerged paper at the next time step and $S_t$ denotes the set of papers published up to current time $t$. The

log-likelihood of the sequence of papers is then simply log $P$. To compare each model against the null model, or a random guesser, we take the log-likelihood-ratio $\log \frac{P}{P(null)}$ which is a standard statistical measure for model comparison. $P(null)$ is theoretically determinable, because at each time $t$ the probability of randomly guessing the true paper to emerge from a pool of $n$ yet-to-emerge papers is $\frac{1}{n}$.

## Results

We evaluated the chaining models described on the historical publication data against a null baseline model that makes random guesses about the emerging order of the publications. For each scientist, we calculated the predictive likelihood of a model for all the yet-to-emerge papers based on Eq 1 aggregated over time (see Materials and methods for details on model evaluation). Fig 2A and 2B show the raw log-likelihood ratios between each model and the null model for the prominent and random-sampled scientists respectively across the 5 fields. Model comparisons against the null indicate the likelihoods obtained in each of the chaining models are significantly higher than those of the random guesser (Mann-Whitney U test: $p < 0.001$ for all conditions; see *Table in* S1 Table for detailed summary statistics). We also examined the relative strength of the models in a winner-take-all analysis, where within each scientist we assessed which model best predicts the emergent pattern of papers over time. We expect the chaining models to perform better than the baseline, and the best model to yield the highest likelihood across scientists. Fig 3A and 3B summarize the results for the groups of prominent and random-sample scientists respectively. *Table in* S2 Table provides the exact model percentages from this analysis. In all cases, the chaining models dominated the null baseline model, and the exemplar-based chaining model yielded the highest predictive likelihood in scientists across fields (binomial test: $p < 0.001$; $n = 10$). Importantly, the same patterns of exemplar advantage were attested in both the group of prominent scientists and the group of random scientists. This initial set of results supports the view that there is temporal continuity in the emergence of scientific ideas with strong statistical tendencies in a shared mechanism of exemplar-based chaining observed robustly across the individuals and scientific fields.

To examine whether collaboration might influence the results described, we next performed the same analysis by controlling explicitly for authorship conditions in the publications. A scientist may play a leading or non-leading role in a co-authored publication, and hence collaboration might introduce randomness in the historical emergence of scientific ideas for any given scientist. To assess whether our results are robust to this factor, we repeated likelihood calculation and winner-take-all analysis on each scientist under 4 different authorship conditions: fewer than 3 authors per paper, fewer than 2 authors per paper, single-author papers, and first-author papers. Here, first-authorship means that the scientist in question is either the first or last author. For each scientist, we included only those papers that meet each authorship condition, and re-ran all analyses. Fig 3 shows that across all 4 authorship conditions, 5 scientific fields, and the groups of prominent and random scientists, the chaining models persistently better predicted the historical data than the null model and exemplar-based chaining is the best overall model (binomial test: $p < 0.0008$; $n = 39$). These results indicate that the regularity underlying the historical emergence of scientific papers is not affected by the degree of collaboration in a scientist's work.

To verify that the best-performing model captures genuine temporal effects, we performed a shuffle test where we randomized the emerging order of papers over the timeline such that the same number of papers emerged at each timestep, but papers were assigned with distorted timestamps. We shuffled 100 times and compared the predominance of the best model in the

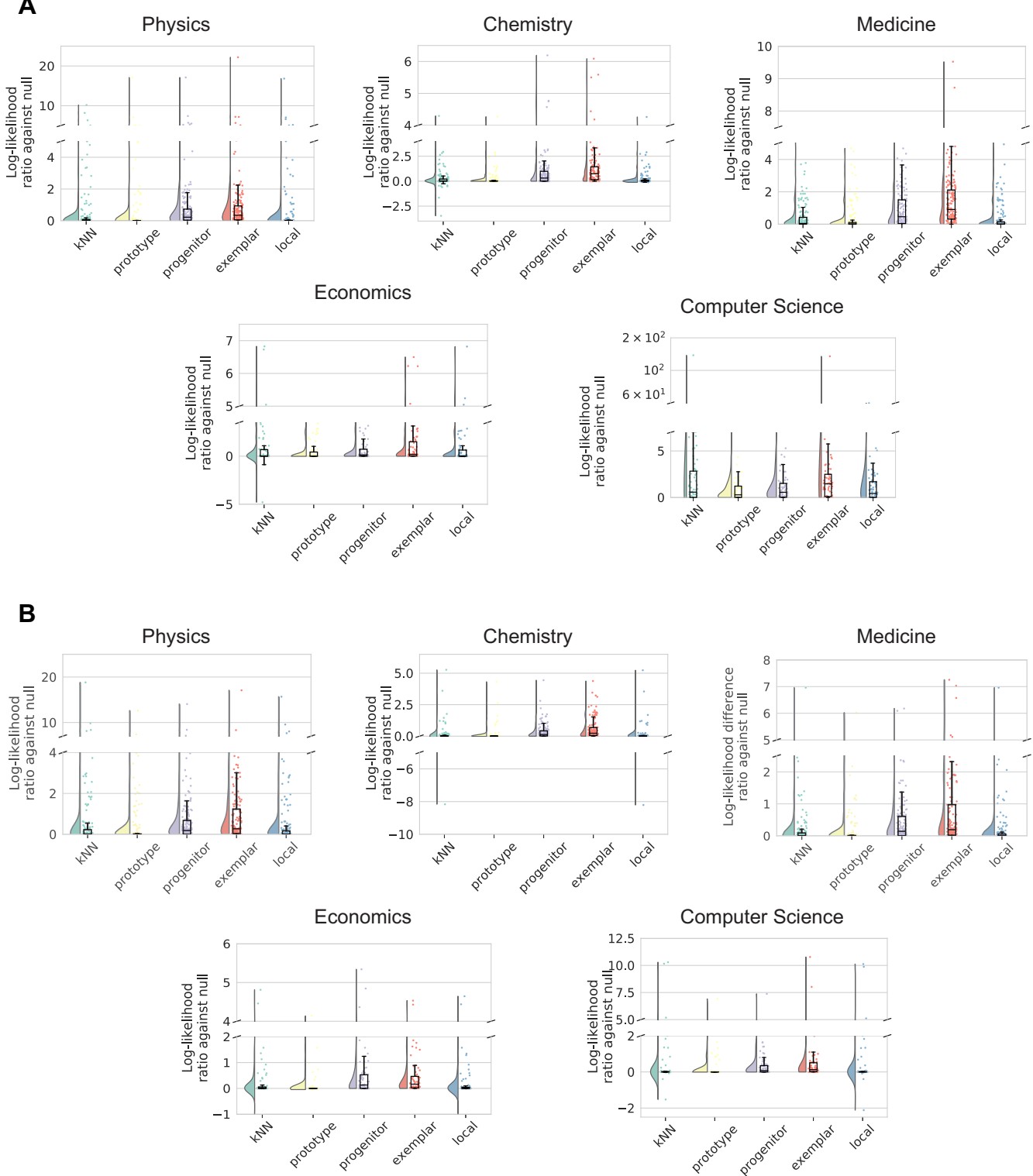

**Fig 2. Summary of model predictive performance.** (A) Log-likelihood ratio of model against the null random baseline model based on predicting historical data from the prominent scientists across the 5 fields. 0 on the vertical axis indicates equal performance between a chaining model and the null model; values above 0 indicate superior performance over the null model. (B) Similar information for the random-sampled scientists.

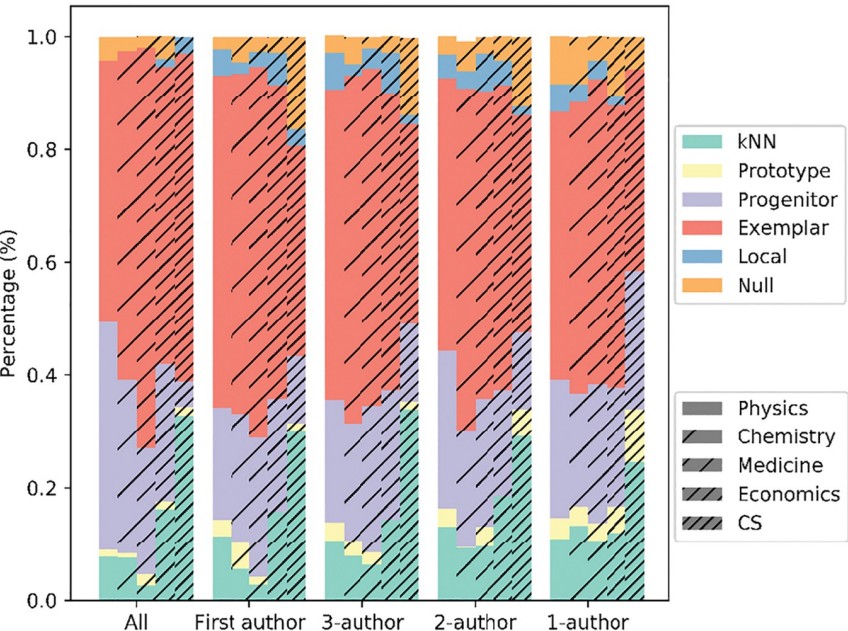

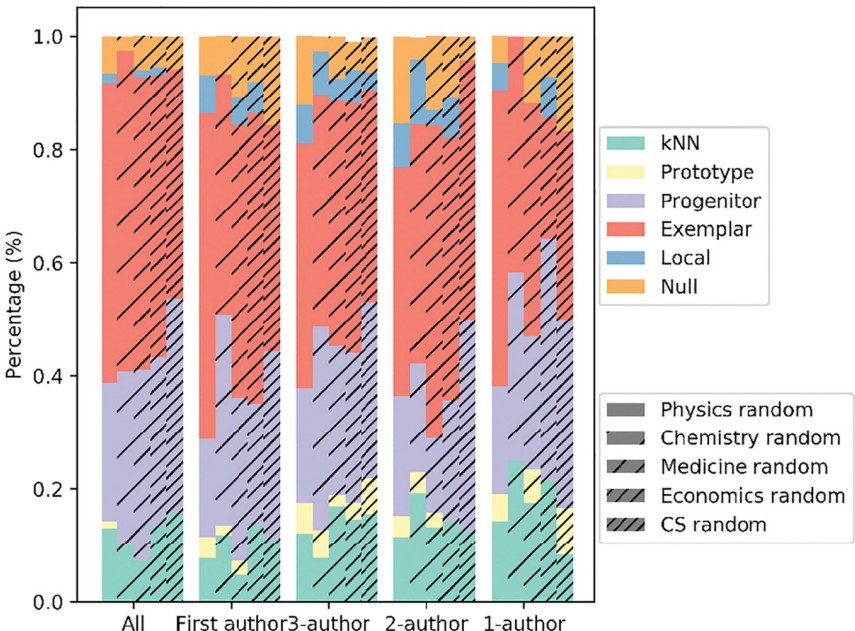

**Fig 3. Summary of results from the winner-take-all analysis.** (A) Bar chart indicates the winning percentage of a model in predicting the historical emergence of publications from the prominent (or award-winning) scientists across the 5 fields ("CS" stands for computer science). Different groups of bar charts display winning percentages under different authorship conditions. (B) Similar information for the random-sampled scientists.

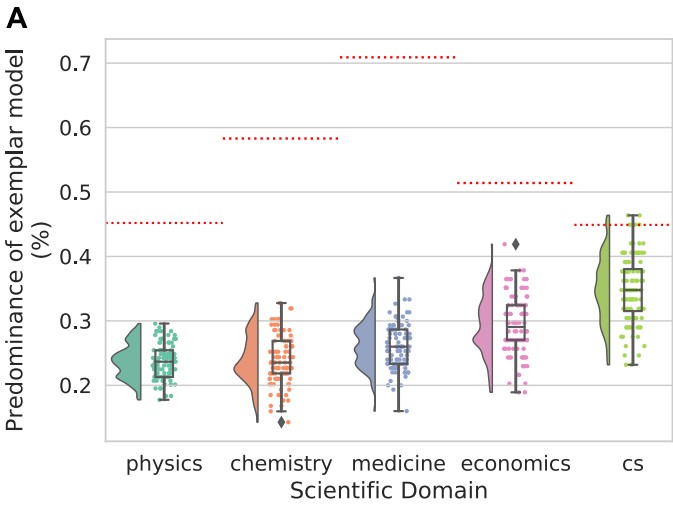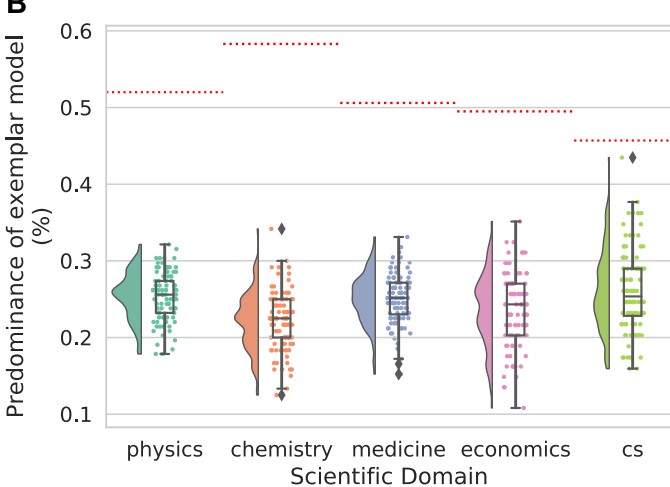

**Fig 4. Summary of results from the time-shuffled analysis.** (A) Comparison of the predominance of the exemplar model (model winning percentage indicated by the red dotted line) against the same statistics recorded in the 100 iterations from the time-shuffled tests (winning percentages shown in the boxplot with an accompanying histogram and scattered raw data). (B) Similar information for the random-sampled scientists.

ground truth timeline to the predominance of the best model in the shuffled cases. If the success of exemplar-based chaining reflects the temporal emergence of scientific papers, the predominance of this chaining model in the ground truth condition should be significantly higher than that in the shuffled conditions. Fig 4A and 4B summarize the results from the shuffle tests for prominent and random scientists across the 5 fields. In all but one case, we found that the dominance of the exemplar model over alternative models is significantly higher in the ground truth condition than in the time-shuffled condition (permutation test: $p < 0.01$ in all cases, except for the prominent computer scientists group $p = 0.06$). These results indicate that the success of the exemplar chaining model reflects the true historical emergence of scientific publications.

Although the exemplar model is predominant through all the predictive analyses, we observed that there is variation among the chaining models in predicting scientists' data. In particular, we found the progenitor model to be persistently the second best-performing model (see Fig 3 and *Table in* S2 Table for the breakdown of model performance). This result shows that despite the fact that exemplar-based chaining may be generally preferred, this preference reflects a statistical tendency and there exist alternative mechanisms in the emergence of scientific publications. For instance, the progenitor model reflects the strategy where a scientist tends to make incremental progress toward a base theme. This model of chaining is statistically less salient than the exemplar-based strategy in our data, but it may be nevertheless adopted quite commonly by scientists. These findings indicate that there are varying strategies in how scientists make progress over time, with exemplar-based strategy being the prevalent kind.

## Discussion and conclusion

Our study offers an initial step toward quantifying and evaluating the regularity in the historical emergence of scientists' outputs across individuals and multiple scientific fields. Our findings reveal a resemblance between the processes of scientific publishing behavior over time and historical word meaning extension: both processes rely on mechanisms of chaining that

grow ideas over time based on closely related existing ideas, which suggests that chaining may serve as a general-purpose mechanism in the extension of human knowledge.

We acknowledge that there are aspects that are not captured by our current analysis and we do not suggest our account to be a full picture of scientists' discovery processes, but rather a high level framework which can be built upon for future work.

One factor influencing scientists' research output that can be further explored is the role of collaboration. Although we analyzed papers written by different numbers of coauthors, we did not examine in detail how the activities of other researchers might impact scientists. This was partly due to data limitations, since we focused on analyzing the sequences of papers and did not consider the collaboration network of scientists. This analysis can be challenging to carry out with the historical Nobel prize winners, since we do not know of any digitized records that would allow us to build a complete collaborative network of scientists from the earlier periods. However, exploring models that take into account collaboration would be an interesting avenue for future research.

The primary focus of our analysis is on examining which of the cognitively plausible models might best account for the historical emergence of scientists' publications. There may be more fine-grained patterns which we are unable to explore here, such as the variation in behavior over the course of career and the variation in behavior due to scientists' differential demographics (e.g., the comparison of senior, more established scientists versus junior scientists).

Throughout our large-scale analysis in multiple scientific fields, we found consistent evidence for exemplar-based chaining in award-winning and random-sampled scientists in their historical emergent patterns of publication. The resolution of our analysis focuses on the level of scientists, so our computational framework does not speak to the emerging properties of individual publications. However, our analysis does reveal that scientists who have done prominent work share similar cognitive strategies in scientific inquiry through time with those who do "normal" science, such that they tend to make progress in continuous ways reflecting a process of chaining that mirrors the processes in historical word meaning extension.

## Supporting information

**S1 Appendix. Validation of semantic representation.** Details of how validation of semantic representations across fields and scientists was performed is included.
(PDF)

**S1 Fig. Validation of semantic representations at field-level.** Validation of semantic representation at the scientific-field level. Comparison of the attested J-value (red dotted line) against the J-values from the shuffled trials across scientific fields (green histogram and boxplot with scattered raw data) in (A) prominent scientists, and (B) random-sampled scientists.
(EPS)

**S2 Fig. Validation of semantic representations at scientist-level.** Validation of semantic representation at the individual-scientist level. Comparison of the attested J-value (red dotted line) against the J-values from the shuffled trials across scientists (green histogram and boxplot with scattered raw data) within each of the 5 scientific fields in (A) prominent scientists, and (B) random-sampled scientists. "CS" stands for computer science.
(EPS)

**S1 Table. Summary statistics from evaluation against the null model, in prominent and randomly-sampled scientists across fields.** Statistics from model evaluation against the null model in log-likelihood ratio for the award-winning scientists as described in the main text, using Mann-Whitney U test. The Mann-Whitney U test was used in place of a t-test because

the data were not normally-distributed (all models outperformed the null, and this is highly significant).
(PDF)

**S2 Table. Percentage breakdowns from winner-take-all analyses.** Tthe percentage break-downs from the winner-take-all analyses described in the main text for the group of award-winning scientists as well as randomly-sampled scientists. The percentage of the model with the highest winning percentage across scientists within each scientific field is marked in bold.
(PDF)

## Author Contributions

**Conceptualization:** Yang Xu.

**Data curation:** Emmy Liu.

**Formal analysis:** Emmy Liu.

**Funding acquisition:** Yang Xu.

**Investigation:** Emmy Liu.

**Methodology:** Emmy Liu, Yang Xu.

**Supervision:** Yang Xu.

**Visualization:** Emmy Liu.

**Writing – original draft:** Emmy Liu.

**Writing – review & editing:** Yang Xu.

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
