## [Decision Letter · Decision Letter 0]

5 Sep 2022

PONE-D-22-15379Regularity in the temporal dynamics of publishing behaviour across scientistsPLOS ONE

Dear Dr. Liu,

Thank you for submitting your manuscript to PLOS ONE. After careful consideration, we feel that it has merit but does not fully meet PLOS ONE’s publication criteria as it currently stands. Therefore, we invite you to submit a revised version of the manuscript that addresses the points raised during the review process.

ACADEMIC EDITOR: Please consider all the comments during the revision and proof read the revised submission.

We look forward to receiving your revised manuscript.

Kind regards,

Qichun Zhang, PhD

Academic Editor

PLOS ONE

Journal Requirements:

Additional Editor Comments :

Some comments have been received from both reviewers and a major revision is helpful to clarify the concerns then enhance the quality of the work.

Reviewers' comments:

Reviewer's Responses to Questions

**Comments to the Author**

1. Is the manuscript technically sound, and do the data support the conclusions?

Reviewer #1: Yes

Reviewer #2: Partly

2. Has the statistical analysis been performed appropriately and rigorously? 

Reviewer #1: Yes

Reviewer #2: No

3. Have the authors made all data underlying the findings in their manuscript fully available?

Reviewer #1: Yes

Reviewer #2: Yes

4. Is the manuscript presented in an intelligible fashion and written in standard English?

Reviewer #1: Yes

Reviewer #2: Yes

5. Review Comments to the Author

Reviewer #1: This is an interesting article highlighting the similarities in the pattern of publishing behaviour of individual scientists over time and the historical word meaning extension. To this end the authors have employed the cognitive theory chaining based on general context model.

The article, though is very well written, it rather very briefly describes the computational framework. It would be better to add more analytical details which would aid the reader to better appreciate the methodology employed.

The article is recommended for publication after the addition of more details concerning model formulation.

Reviewer #2: The paper investigated an important 'social behaviour' using computing technologies. Basically, the topic is interesting for the academics. However, some issues should be taken into account during a major revision.

1) The motivation is not clear, most of the scientific findings are given based on one single amazing idea and the original idea cannot be predicted. So why we need this statistical research work?

2) Various methods have been used in the manuscript such as clustering method KNN, why these methods have been selected? Is the comparative study the main contribution?

3) In figure 1, yellow dot data has been marked as 2025. Where is this data from? I understand that the work is based on historical data.

6. PLOS authors have the option to publish the peer review history of their article (what does this mean?). If published, this will include your full peer review and any attached files.

Reviewer #1: No

Reviewer #2: No

---

## [Author Response · Author response to Decision Letter 0]

18 Oct 2022

=== Reviewer 1 ===

We appreciate your review and recommendations on our work. We have revised the manuscript to motivate the computational framework better, and to make the formulation of models clearer.

> It would be better to add more analytical details which would aid the reader to better appreciate the methodology employed.

Thank you for your feedback, we have added further description to paragraph 1 of the ``Computational Framework'' section, in which we clarify our methodology and framing. We have also added additional details about SBERT, which is the embedding model we used, in the same section.

> The article is recommended for publication after the addition of more details concerning model formulation.

Thank you, we have added exposition to paragraph 5 of the ``Computational Framework'' section. In particular, each of the models examined is specified completely by Table 1 and Equation 1. We have made this more explicit in the manuscript in paragraph 11 of this section. 

=== Reviewer 2 ===

We would like to thank you for your feedback. We clarify our methods and motivation behind the paper: we are not seeking to predict what the amazing ideas that actually emerge will be, but rather trying to characterize the overall process that scientists use to extend their ideas. We have also added more detail to the "Computational Framework" section to make the motivation more clear to readers. 

> The motivation is not clear, most of the scientific findings are given based on one single amazing idea and the original idea cannot be predicted. So why we need this statistical research work?

Thank you for this clarification question. We wish to emphasize that the purpose of the paper is not to predict what amazing single scientific ideas will arise, but rather to characterize the overall chaining process in a scientist's publications as they emerge through time. Novel scientific ideas have some relationship to the existing scientific literature and a scientist's past work, and it's this relationship between past work and future work that motivates our analysis with the temporal chaining models. This is similar to the word-sense extension models mentioned in paragraph 6 of the Introduction: we cannot predict exactly what novel sense(s) of a word will arise next, but we can characterize the process of sense extension and hence making predictions about the emerging order of senses over time. Similarly, we can characterize the overall growth of a scientist's publications over time, even if we cannot predict exactly what that scientist will do in the future.

> Various methods have been used in the manuscript such as clustering method KNN, why these methods have been selected? Is the comparative study the main contribution?

We have added exposition and motivation to the ``Computational Framework'' section of the manuscript. These methods were selected based on past work in word-sense extension, and comparative study is the main contribution, as all of these extension mechanisms are cognitively plausible and based on psychological research. Another contribution is that we find the best model to be the exemplar model, which mirrors the findings in word-sense extension, and suggests that this may be a natural and general way in which people extend concepts.

> In figure 1, yellow dot data has been marked as 2025. Where is this data from? I understand that the work is based on historical data.

We have data up until 2020, and the plot has now been fixed, since it was caused by matplotlib's automatic label generation.

---

## [Decision Letter · Decision Letter 1]

16 Nov 2022

Chaining and the temporal dynamics of scientists' publishing behaviour

PONE-D-22-15379R1

Dear Dr. Liu,

We’re pleased to inform you that your manuscript has been judged scientifically suitable for publication and will be formally accepted for publication once it meets all outstanding technical requirements.

Kind regards,

Qichun Zhang, PhD

Academic Editor

PLOS ONE

Additional Editor Comments:

All the concerns have been addressed well in the revised version and both reviewers satisfy the revision. Therefore, I recommend accepting this manuscript as it is.

Reviewers' comments:

Reviewer's Responses to Questions

**Comments to the Author**

1. If the authors have adequately addressed your comments raised in a previous round of review and you feel that this manuscript is now acceptable for publication, you may indicate that here to bypass the “Comments to the Author” section, enter your conflict of interest statement in the “Confidential to Editor” section, and submit your "Accept" recommendation.

Reviewer #1: All comments have been addressed

Reviewer #2: (No Response)

2. Is the manuscript technically sound, and do the data support the conclusions?

Reviewer #1: Yes

Reviewer #2: (No Response)

3. Has the statistical analysis been performed appropriately and rigorously? 

Reviewer #1: Yes

Reviewer #2: (No Response)

4. Have the authors made all data underlying the findings in their manuscript fully available?

Reviewer #1: Yes

Reviewer #2: (No Response)

5. Is the manuscript presented in an intelligible fashion and written in standard English?

Reviewer #1: Yes

Reviewer #2: (No Response)

6. Review Comments to the Author

Reviewer #1: The authors have carried out the revision by adding a paragraph highlighting the analytical details as required and the paper is accepted.

Reviewer #2: All my concerns have been addressed well in the revised version. Thus I think it is ready for publication.

7. PLOS authors have the option to publish the peer review history of their article (what does this mean?). If published, this will include your full peer review and any attached files.

Reviewer #1: No

Reviewer #2: No

---

## [Editor Report · Acceptance letter]

16 Dec 2022

PONE-D-22-15379R1 

Chaining and the temporal dynamics of scientists’ publishing behaviour 

Dear Dr. Liu:

I'm pleased to inform you that your manuscript has been deemed suitable for publication in PLOS ONE. Congratulations! Your manuscript is now with our production department. 

Kind regards, 

on behalf of

Dr. Qichun Zhang 

Academic Editor

PLOS ONE